# A Graded Approach for Evaluating Health Claims about Plant-Based Food Supplements: Application of a Case Study Methodology

**DOI:** 10.3390/nu13082684

**Published:** 2021-08-02

**Authors:** Hélène Chevallier, Florent Herpin, Hélène Kergosien, Gabrielle Ventura, François-André Allaert

**Affiliations:** 1Biofortis, 44800 Saint-Herblain, France; 2CEN Nutriment, 21000 Dijon, France; contact@groupecen.com; 3Synadiet, 75009 Paris, France; hkergosien@wanadoo.fr (H.K.); nouscontacter@synadiet.org (G.V.)

**Keywords:** health claim, food supplement, nutrition regulation

## Abstract

The implementation of REGULATION (EC) No 1924/2006 has led to the formation of a list of health claims that can be used in food supplements (EU 432/2012). However, such supplements are often composed of plant preparations with claims omitted from this list. The peculiarity of plants is related to their long history of use, that could allow claims based on traditionally recognized health effects. In addition, the scientific literature has been enriched over the years through clinical studies that have assessed the bioavailability and efficacy of bioactive components, and investigated their mechanisms of action. Based on existing recognized models which aim to classify research according to the level of scientific evidence, Synadiet developed a three-grade model (A, B or C) for assessing plants health claims. In this paper, the applicability of the model is illustrated through an example for which a Grade B health claim attesting the possible contribution of red clover isoflavones to the improvement of blood lipid levels in postmenopausal women has been attributed. The model appears able to be easily extrapolated to claims pertaining to other plants. If adopted by consensus at European level, this model could initiate the implementation of a positive list of health claims on plant preparations.

## 1. Introduction

When it comes to substantiating claims about human health, there can be no question that the highest possible standard of proof is required. However, regarding health claims of food supplements related to plants, to date, there is a status quo concerning their evaluation (pending authorization). This does not provide a satisfactory solution either for consumers or for the industry. The intent of the work performed by Synadiet was to propose a model based on graded claims, where the level of proof provided is evaluated by a standardized methodology and where traditions of using plant-based products are taken into account. This attempts to conciliate the requirements of European public health authorities and the concerns of the industry, while keeping in mind consumer protection in terms of provided information.

### 1.1. Context of Plant-Based Food Supplements

Plants have been utilized since time immemorial for the relief of human illnesses. Many have been used in allopathic drugs or as phytomedicines and have a long history of use. This is the case, for example, for *Aesculus hippocastanum* L. [1] in venous disease, *Ginkgo biloba* L. for cognitive disorders [2], *Pygeum africanum Hook f.* in benign prostate hyperplasia [3], etc. Notably, however, the same plant may have different statuses (as a drug or food supplement) depending on the dose, the intended use, and the supposed benefit for human health or well-being. It is also important to specify that, for the same plant, a multitude of plant preparations exists, depending on the part of plant used and the preparation process; this needs to be considered. In addition, several physiological benefits may be related to the same plant. Some vitamins and minerals have obtained claims on the basis of full substantiation data, others have been approved on the basis of consensus of expert opinion (vitamin C and function of the immune system, or zinc and cognitive function, or vitamin A and the metabolism of iron) [4]. The same rules could be applied to generic functional claims for plants where there is documented long-lasting use compatible with the area of the application of supposed health benefits. As recommended by Synadiet, and according to the publication of Anton et al., evaluations should be conducted on the basis of official monographs (such as those of the EMEA, WHO, Health Canada, etc.) and reference text books on the properties of plants for which documented uses for at least 25 years exist, representing the equivalent of one generation [5,6]. The volume of existing recognized documentation should make it possible to rapidly issue a list of traditional claims for a number of plant preparations. Indeed, for now, the model for evaluating (by EFSA) and authorizing claims (by the European Commission) does not allow consideration of the tradition of use. In addition, a substantial “pending list of health claims” related to plants is currently being used. These claims were submitted during the implementation of the REGULATION (EC) No 1924/2006 but not assessed yet due to the lack of adequate methodology that takes into account their specificity. It is time to offer an alternative, both to supervise communication towards consumers and also to support the development of research at industrial level.

### 1.2. The Graded Health Claim System: Principles

To be able to simultaneously take into account traditional effects and also new evidence from clinical studies that have been, and continue to be, conducted on plants, it seems desirable to shift towards a graded evaluation system, such as the U.S. FDA Qualified Claims [7]. This view of ranked standards of proof has also been adopted into the recommendations or consensus meetings of the French authorities (HAS/ANSES) or international medical societies such as the European Society of Hypertension (ESH) [8]. As developed in a previous paper, the idea of grading claims according to the strength of the available evidences, as is conducted for the guidelines/recommendations, is a very good principle and is more informative for consumers than the current dichotomic practice, which can be summarized as certainty or nothing [9].

Therefore, Synadiet has formulated a proposal for a graded health claims system for plant-based products. This model, presented in Table 1, was developed according to pre-existing models, presented in Table 2 and Table 3, which represent the WHO guidelines for methodologies on the research and evaluation of traditional medicine and its associated grade system for the level of evidence according to the type of data. In the system proposed for plant food supplements by Synadiet, in order to enable transparency for consumers, claims are graded in three levels: A, B, and C. Level C corresponds to “*traditionally used for*” and does not raise wording difficulties (its definition will be given hereafter), contrary to level A and B claims which require clear and precise wording, supported by the appropriate strength of evidence to substantiate them.

With regard to Grade A, in Table 2, the corresponding Level Ia, “Evidence from a meta-analysis of randomized controlled trials”, apart from very rare exceptions, does not exist today for plant-based food supplements; Level Ib is the standard which must be adopted for supporting the grading health claim A: “Evidence from at least one randomized controlled trial”. According to the grade position, it is important to emphasize that to achieve Grade A, two converging randomized controlled trials (RCTs) are not mandatory, and a single qualitative study may be enough. It is expected that such a Level Ib RCT will exhibit a high-quality design in respect to risks of bias, and will need to demonstrate statistical as well as clinical significance of the observed effect. It should be noted that the effect of the product needs to be observed compared to a placebo or a comparator, and not according to statistical analysis, i.e., “before/after”. Therefore, the proposal of Synadiet for “Two convergent studies with at least one human RCT of Level Ib” appears a good compromise between EFSA’s current requirements and the Grade conditions. However, it is important to emphasize that a Level Ib human RCT should be conducted on a well-characterized plant preparation, which is unfortunately not the case in many published clinical trials. Other converging human studies may be controlled studies without randomization (Level IIa), other quasi-experimental studies (Level IIb), or observational studies (Level III). Their definitions are given in Appendix A. There may also be supporting evidence such as animal and in vitro studies, which provide more information surrounding the relevant mechanisms of action, or consumer satisfaction tests, without being sufficient to substantiate a health claim alone.

With regard to Grade B, Synadiet proposes the following level of corresponding evidence: “Convergent body of evidence including at least one randomized controlled study (Level Ib) that does not fulfil the quality criteria established for Grade A, or one controlled study without randomization (quasi-experimental study) (Level IIa), or one other type of quasi-experimental (non-controlled) study (Level IIb) and eventually evidence from observational studies (Level III)”. This definition has the advantage of paying attention to observational studies which reflect the use of plant-based products in real life and the fact that many such substances already have a long history of daily use and were commercialized before today’s requirements for proving health claims. Additionally, clinical textbooks can be provided, which describe how constituents identified in plant preparations work in the body, supporting evidence such as animal and in vitro studies that provide more information surrounding the mechanism of action and consumer satisfaction tests. Grade B can be sustained by several studies that constitute the body of evidence.

On that basis, the wording for Grade B recommendations would be, for example, “can contribute to the decrease in xxx” or “can contribute to the improvement of xxx”; Grade A wording would be “decreases xxx”, “relieves xxx” or “improves xxx”.

With regard to Grade C, “traditionally used for xxx”, it is based on authoritative reference texts corresponding to Level IV evidence, defined as “Evidence from authoritative reference textbooks, scientific opinion from scientific organizations, regulatory authorities and documented history of uses, e.g., classical texts, published documents from scholars or experts that report the traditional use of the ingredient concerned”. Traditional use is restricted to products for which a long-lasting use of at least 25 years can be documented.

Each type of graded claim will be related to the parts of the plant, the type of preparation and the conditions of use.

## 2. Materials and Methods

To illustrate the applicability of the model described above, a case study has been performed regarding the assessment of the pending health claim related to *Trifolium pratense* (red clover) and cardiovascular health. The use of systematic reviews as a standardized method to identify, select and critically appraise relevant research enables the synthesis of the current body of knowledge on targeted plant efficiency and safety issues. A systematic review was conducted to apply the graded evidence claim methodology according to the EFSA Guidance [10] based on eight core steps (Figure 1).

This method was applied to determine whether the pending health claim related to *Trifolium pratense* (red clover) and cardiovascular health is justified (ID 2496) [11]. The proposed wording for the actual pending claim is: “*Trifolium pratense* is beneficial for the cardiovascular system”, with the following conditions of use: *“1000–2000 mg of red clover in tablet or capsule form for 6 weeks”*. Two types of bibliographic references were used for this study. On the one hand, references were addressed to EFSA in 2008 to support the claim (consolidated list of Article 13 health claim—list of references received by EFSA [11]). On the other hand, a literature search was performed to highlight subsequent publications in the period from 2008 to 2017 related to the pending claim. Indeed, in order to develop a consistent list of references, a literature search was performed in December 2017 on PubMed, Science Direct, and Google Scholar. The objective of the literature search was to identify human, animal, and in vitro studies that evaluated the impact of supplementation with isoflavones of red clover (in the form of titrated extract or purified isoflavones) on cardiovascular health. Meta-analyses, reviews, intervention studies, and observational studies were selected. The following keywords were used: (“red clover” OR “trifolium pratense”) AND (“cardiovascular” OR “cholesterol” OR “triglycerides” OR “blood pressure” OR “blood lipids”) AND (“oral” OR “dietary” OR “supplementation”).

Sixteen studies were initially referenced for EFSA in 2008 in order to substantiate the claim [11]. Among these, it was decided to exclude three studies only available in Latvian from the review, because no analysis could be conducted. In addition, an in vivo review was also excluded due to the summarization of articles already present in the list (it presented no additional data). Three studies were dedicated to a mixture of red clover isoflavones and soy, or to a mixture of red clover and probiotics, did not find any significant beneficial effects on lipemia or other cardiovascular risk factors [12,13,14]. These studies did not provide any additional information regarding the effects of red clover isoflavones on cardiovascular parameters; therefore, they were also discarded from the evaluation process of substantiating the claim. However, these studies were taken into consideration for the conclusion of the grade of the health claims. Consequently, among the sixteen studies listed for EFSA pending claims, nine were selected for the justification of the claim. Table 4 presents the study classifications and exclusion statuses.

Regarding the literature search from 2008 to 2017, ten studies were highlighted. Three studies conducted in mouse models were considered not relevant for the justification of the claim, because they were performed on animal disease models (type II diabetes and hepatic steatosis models). In addition, three clinical studies where no significant beneficial effect was observed on cardiovascular system parameters, and one clinical study conducted to assess the association of red clover isoflavones with other substances (probiotics) were also excluded, because they could not contribute to the justification of the claim. However, these studies were taken into consideration for the conclusion of the grade of the health claim. As a result, among the ten studies, four were selected for substantiation of the claim (Table 4).

To summarize, according to all the data available in the literature, thirteen studies were analyzed in order to investigate whether the pending health claim relative to *Trifolium pratense* (red clover) and cardiovascular health is justified. The diagram in Figure 2 summarizes the identified data, and Table 5 presents the characteristics of these studies.

Among the thirteen clinical studies where beneficial health effects were highlighted, some were considered as more relevant regarding the justification of a claim, even if all contributed to the level of evidence. It should also be noted that among the thirteen publications, four publications actually referred to only two distinct clinical trials: [22,24] refer to only one study and [18,28] refer also only to one study.

## 3. Results

### Study Characteristics

Regarding the study populations, all investigated the effects on menopausal women, one additionally included pre-menopausal women [15], and two also included men [22,24]. The effects were mainly observed in postmenopausal women, excepted for Nestel et al., who only observed effects in men [22]. The study from Howes et al. was considered less relevant for substantiating a claim for the general population, because the study sample was composed of menopausal women with type 2 diabetes [14]. In three studies, the product was not compared to a placebo but to a control group without supplementation or a placebo [16,32,33]. The selected studies mainly evaluated the effect of a supplementation on the evolution of blood lipids (total cholesterol, LDL cholesterol, HDL cholesterol, triglycerides) as well as arterial pressure and elasticity of the arteries. Only four studies among the thirteen specified criteria being cardiovascular parameters [20,22,24,29]; the others were not related or unspecified in the article. The doses evaluated in the studies presented in this synthesis are indicated in “*isoflavones*” equivalent and were between 40 mg/day and 80 mg/day. The duration of investigations was between one month and eighteen months. We also noticed that in the majority of the cases, no sample size calculation was specified. Regarding statistical results, ten studies showed statistically significant results between groups, which is a relevant level of evidence [14,15,18,20,21,22,23,24,32,33]. For these studies, we analyzed the number and the characteristics of subjects considered in the analysis. As an example, studies which observed significant differences only for a sub-population of the entire analysis (i.e., only men for Nestel et al.; or only women with BMI ≥ 25 kg/m^2^ for Chedraui et al.) and studies with a very small sample size were considered with caution during discussion [14,15,21]. Indeed, statistical considerations have a key place in the analysis for assessing the relevance of the results. Sample size calculation and subgroup analysis (predefined or post-hoc analyses, type of comparisons, etc.) need to be well identified and taken into consideration during the evaluation process. Finally, seven studies highlighted interesting results in terms of clinical relevance for the justification of the claim; Table 5 presents the detailed characteristics and outcomes of the most relevant studies regarding the pending claim for *Trifolium pratense* [15,20,24,29,32,33].

It seems difficult to confirm the proposed claim “*Trifolium pratense is beneficial for the cardiovascular system*” as it stands, based on the studies reviewed. Indeed, these were generally conducted on a small sample size and on populations mainly composed of menopausal women (which is only a subgroup of the general population). Conclusions should therefore consider that the active ingredient has not been tested in the general population. Consequently, claims of health benefits should be worded focusing on postmenopausal women population, who can be considered at risk because of hormonal imbalances due to their physiological situation. It would also be interesting and possible to test the effects of red clover supplementation on a population with hypercholesterolemia/triglyceridemia/hypertension, if they do not require a prescribed drug management regime. Indeed, it would be relevant to target this population (biological limits to be defined) to be able to observe physiological effects. A randomized, placebo-controlled clinical study in a population characterized as “*at risk*” based on their blood status and assessing changes in lipid parameters could be interesting.

Moreover, the studies were heterogeneous in terms of the evaluated parameters, making the comparison and correlation of studies difficult. The “*cardiovascular health*” axis refers to several physiological effects in the cardiovascular system; therefore, there are many criteria for assessing cardiovascular health. Indeed, the EFSA’s guidance allows claims related to changes in the blood lipid profile, reductions in blood pressure, endothelial function, reduced platelet aggregation, or homocysteine for cardiovascular health [35]. Therefore, it seems important to consider a specific physiological effect rather than the “*cardiovascular health*” axis in general. Regarding the doses and duration of supplementation, the doses proposed for the health claim labelling submission were “*1000–2000 mg of clover*”, without specifying the composition of the extract. Notably, in France, “*l’arrêté plante*” recommends a maximum consumption of 1 mg/kg of body weight per day in “*aglycone*” equivalent. The doses tested were also heterogeneous between studies, making the drawing of a general conclusion difficult. Regarding the chemical forms used (biochanin or formononetin, for example), they were not always specified in the publications; therefore, we can question the bioavailability and efficacy of these different plant polyphenols. Moreover, it would be more appropriate to express the dose in “*red clover isoflavones*” rather than “*red clover*” in the condition of use. Regarding the duration of investigations, they were consistent with EFSA guidance requirements (short-term effects observed over three to four weeks, then long-term effects observed up to eight weeks), as well as with pending claims (six weeks of consumption).

In the assessment, it is also important to consider the strength of the evidence where some studies have failed to present any significant effects on cardiovascular parameters. This may be the consequence of an absence of an effect or a low statistical power (there were no calculations of sample size for the majority of trials). A meta-analysis would increase this power, including the difficulty of finding a common criterion relevant to all studies. In addition, it should be noted that cardiovascular parameters were presented as the main criterion for only one study [22], and for two others as a composite criterion, meaning that the main criterion was a composition of different criteria (BMD and LDL-c [29] and HDL-c, osteocalcin, and urinary N-telopeptide [20]).

The results of these studies do not enable a consensus to be formed as to the overall effectiveness of red clover on the cardiovascular system (i.e., an improvement in one parameter does not necessarily allow generalization of a beneficial effect on the entire cardiovascular system). Clinical studies are needed to investigate the effect of this plant extract on cardiovascular health by focusing on specific parameters, such as the concentrations of LDL-c, HDL-c, and triglycerides, which would be able to provide beneficial clinical data for the consumer. In addition, six studies of level Ib did not show any significant results on cardiovascular parameters, and thus were also taken into consideration when assessing the grade of this claim.

In conclusion to this study case, the body of evidence mainly constitutes “Ib-type” studies, which could lead to a Grade B claim that *Trifolium pratense* “can contribute to” certain health benefits. The wording of the claim may be: “*Isoflavones of red clover can contribute to lowering LDL-cholesterol in postmenopausal women, with a proposed dose of 50–80 mg/day of red clover isoflavones. Lowering LDL-cholesterol helps reduce the risk of cardiovascular disease*”. This is with the support of studies from Clifton-Bligh et al. (2015), M. Terzic et al. (2012), M. M. Terzic et al. (2009), and Chedraui et al. (2008). This grading system has taken into account the evaluation criteria, the studies’ populations, the conditions of use of the product, and the relevance of the statistical data.

## 4. Discussion

The Grade system enables the classification of publications by the level of evidence according to their designs (randomized, controlled, double-blind, observational study, etc.). However, critical reading is necessary for assessing the relevance of the results obtained and the conclusions of a study: study design, suitability of primary and secondary outcomes, sample size calculation, population characteristics set analysis (intention to treat, per-protocol), statistical analyses considering identification and control of potent bias, etc. Consequently, Synadiet’s model also takes into account overall parameters related to the quality of the results.

Regarding the current health claim regulations for food supplements, there is a risk that the absence of information that arises from an all-or-nothing evaluation could lead to consumers being given false information or even being misled about many products (looking up inaccurate information on websites, misinterpreting certain health messages, etc.). The current system, whereby 80% of products do not carry health claims, exposes consumers to the continued absence of any advice about the products for which there are no claims. Considering this situation, it seems that an evaluation able to consider the levels of evidence appears to be a good solution from a scientific and regulatory point of view. This is the direction in which the U.S. Food and Drug Administration has been moving for years [7]. Our suggestion for future developments of the presented graded evidence methodology appears to be easily applicable and could better inform subjects, rather than depriving them from all forms of information which did not achieve a Grade A level.

Grading health claims on plant food supplements according to the strength of evidence of the clinical studies conducted is not easy because all graded systems are linked to a predominant position attributed to the RCT, as well as to a level of scientific requirement that includes not only good-quality RCTs, but also meta-analyses or systematic reviews. The major difficulties in highlighting systematic reviews or meta-analyses on plant-based supplements are that: (1) clinical studies for food supplements are not mandatory for market declarations and are therefore less frequently conducted than for drugs; and (2) the heterogeneity of trial data characteristics has increased the difficulty of pooling clinical trials; for example, extracts are not always well defined and reported in studies, doses and populations are not the same, etc. In addition to this, the high level of evidence based on meta-analysis or on systematic review of RCTs are not even required for the market authorization of drugs and therefore there is no reason why they should be required for plant food supplements. Moreover, during the conduct of the study case, it appeared necessary to add some requirements to the Synadiet’s grading system, especially on the level of evidence and methodological aspects of the study.

The case study conducted on *Trifolium pratense* shows, firstly, that the methodology is applicable and efficient to lead to a conclusion based on convincing arguments. Secondly, our graded approach also highlights that each health claim needs to be analyzed case by case by a careful literature review. It also demonstrates that a graded system with an intermediary position, which is Grade B, “*can contribute to*”, can be interesting to enhance the existing scientific evidence, with distinctions between the levels of proof. Indeed, Grades A, B and C can then be easily discriminated, because Grade A use action verbs, and Grade B uses the conditional, “*can*”. In addition, traditional uses are taken into account with Grade C, which can be used with proper documentation. To accurately assess the reliability of the model, Synadiet conducted several other case studies on pending botanical health claims and concluded that “*a minimum of 120 mg Bacosides from a Bacopa monnieri extract standardized in Bacosides (USP method)*
*improves cognitive faculties and memory*” (Grade A) when assessed on healthy subjects over a period of twelve weeks; and *“Harpagophytum procumbens L. is traditionally used for maintaining joint health”* (Grade C). It was also concluded that the available data were not sufficient to substantiate any claims regarding the benefit of *Plantago lanceolata L.* on the mobility and flexibility of joints.

It should be noted that this case study also raised different requirements for labelling and assessing a claim; the targeted claims need to be fully identified in terms of population, conditions of use, and assessment parameters (evaluation criteria should be relevant for the axis and be coherent between the different clinical studies). Statistical analysis reliability is also taken into account.

## 5. Conclusions

Our graded approach for evaluating health claims about plant-based food supplements provides reliable information to consumers, in contrast to “all-or-nothing information”. This graded approach is, to date, the most highly developed which fits with the purposes of the Regulation (EC) No 1924/2006 of the European Parliament and of the Council of 20 December 2006 on nutrition and health claims made on foods, which were clearly implemented to protect the consumer. As presented, clinical studies play a key role in the evaluation of health claims (for all types of products). More than a tool to substantiate current health claims, this model of graded health claims will stimulate food supplement developers to invest in clinical research.

## Figures and Tables

**Figure 1 nutrients-13-02684-f001:**
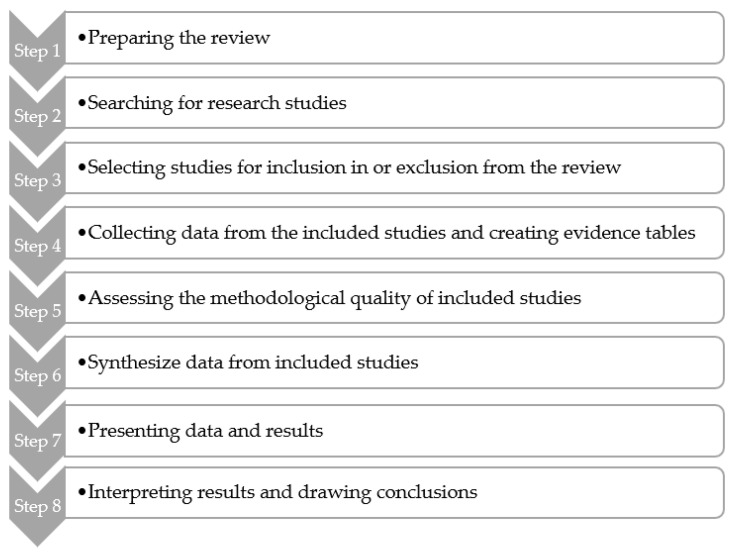
Core steps for performing a systematic review (adapted from the Cochrane Handbook for Systematic Reviews of Interventions, Higgins and Green (editors), 2009, and from the EFSA Guidance for those carrying out systematic reviews (*EFSA Journal*, 2010; 8(6):1637)).

**Figure 2 nutrients-13-02684-f002:**
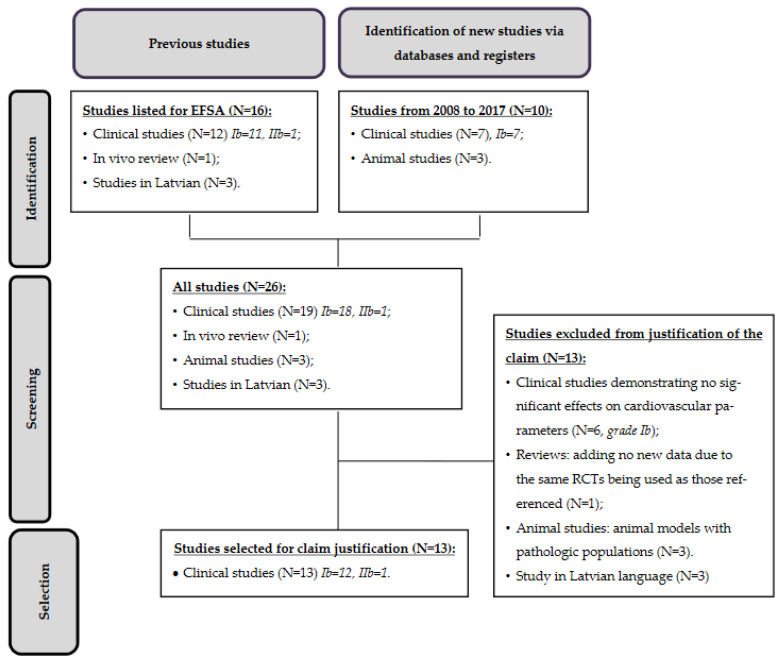
Flow diagram of study selection regarding the pending claim.

**Table 1 nutrients-13-02684-t001:** Synadiet’s model proposal for botanical health claim evaluations.

Grade	C	B	A
	Traditionally Used for…	Can Contribute to…	Decreases... Relieves… or Improves…
General health claim “General function” claims refer to the role of a nutrient or substance in growth, development, body functions, biological activities, body structure, psychological and behavioral functions, weight control to maintain good health, and comfort. Reduction in disease risk factors Any health claim that states, suggests or implies that the consumption of a food category, a food, or one of its constituents significantly reduces a risk factor in the development of a human disease, including for example: Cholesterol, triglycerides/Blood pressure; Bone mineral density/Osteoporosis; Blood glucose; Macular density/vision; Bacterial adhesion/urinary tract infections.	Authoritative reference texts: Reference textbooks (Pharmacopoeias, monographs and scientific journals); Scientific opinion from scientific organizations or from regulatory authorities; Documented history of use, e.g., classical texts, published document from scholars or experts that report the traditional use of the ingredient. Dated a minimum of 25 years ago	Convergent body of evidence with at least: One randomized controlled study (Level Ib) with bias or One controlled study without randomization (quasi-experimental study) (Level IIa) or One other type of Quasi-experimental study (Level IIb) AND Observational study (Level III) and/or Clinical textbooks that describe how constituents determined in the plant preparation work in the body and/or Supporting evidence such as animal and in vitro studies that provide more information surrounding the mechanism of action and/or Consumer satisfaction tests (visual analogue tests, Likert scales)/quality of life questionnaire Very relevant company-owned scientific data (published and unpublished) can be submitted, if available	Two convergent studies with at least: One randomized controlled study (Level Ib) on a characterized plant preparation and Human studies: controlled study without randomization, other type of quasi-experimental study, observational studies, etc. (Level IIa, IIb, III) or Supporting evidence such as animal or in vitro studies that provide more information surrounding the mechanism of action or Consumer satisfaction tests (visual analogue tests, Likert scales)/quality of life questionnaire Very relevant company-owned scientific data (published and unpublished) can be submitted, if available

**Table 2 nutrients-13-02684-t002:** Grading of evidence and recommendations according to the grade system.

Grade	Type of Evidence
Ia	Evidence from a meta-analysis of randomized controlled trials
Ib	Evidence from at least one randomized controlled trial
IIa	Evidence from at least one controlled study without randomization
IIb	Evidence from at least one other type of quasi-experimental study
III	Evidence from observational studies
IV	Evidence from expert committee reports or experts

**Table 3 nutrients-13-02684-t003:** General Guidelines for Methodologies on Research and Evaluation of Traditional Medicine WHO/EDM/TRM2000.1 annex IV.

Grade	Recommendation
A (Evidence levels quality Ia, Ib)	Requires at least one randomized controlled trial as part of the body of literature of overall value and consistency addressing the specific recommendation.
B (Evidence levels IIa, IIb, III)	Requires the availability of well-conducted clinical studies but no randomized clinical trials on the topic of recommendation.
C (Evidence Level IV)	Requires evidence from expert committee reports or opinions and/or clinical experience of respected authorities. Indicates an absence of directly applicable studies of good quality.

**Table 4 nutrients-13-02684-t004:** Studies classification and exclusion status.

Author_Date	Type of Evidence Classification	Inclusion or Exclusion in the Analysis in Favor of the Pending Claim
**Studies from EFSA’s list**		
Atkinson_2004 [12]	Ib	Excluded, no significant results on cardiovascular parameters
Blakesmith_2003 [13]	Ib	Excluded, no significant results on cardiovascular parameters
Campbell_2004 [15]	Ib	Included
Clifton-Bligh_2001 [16]	IIb	Included
Geller_2006 [17]	Review	Excluded, utilization of articles already present in the analysis
Hidalgo_2005 [18]	Ib	Included
Howes_2000 [19]	Ib	Excluded, no significant results on cardiovascular parameters
Howes_2003 [14]	Ib	Included
Knudson Shult_2004 [20]	Ib	Included
Nestel_1999 [21]	Ib	Included
Nestel_2004 [22]	Ib	Included
Rubine_2001	/	Excluded, Latvian language only, classification not possible. Not referenced on classical database.
Rubine_2004	/	Excluded, Latvian language only, classification not possible. Not referenced on classical database.
Skutelis_2005	/	Excluded, Latvian language only, classification not possible. Not referenced on classical database.
Squadrito_2002 [23]	Ib	Included
Teede_2003 [24]	Ib	Included
**Studies from the literature search for the period 2008–2017**		
Chen_2014 [25]	Animal study	Excluded, animal models not relevant (disease)
Qiu_2012a [26]	Animal study	Excluded, animal models not relevant (disease)
Qiu_2012b [27]	Animal study	Excluded, animal models not relevant (disease)
Chedraui_2008 [28]	Ib	Included
Clifton-Bligh_2015 [29]	Ib	Included
Lambert_2007 [30]	Ib	Excluded, no significant results on cardiovascular parameters
Mainini_2013 [31]	Ib	Excluded, no significant results on cardiovascular parameters
Terzic_2009 [32]	Ib	Included
Terzic_2012 [33]	Ib	Included
Thorup_2015 [34]	Ib	Excluded, no significant results on cardiovascular parameters

**Table 5 nutrients-13-02684-t005:** Characteristics of the most relevant studies regarding the pending claim for *Trifolium pratense*.

Authors	Number of Subjects	Population	Primary Endpoint	Evaluation Criteria of the Cardiovascular System (Observed Significant Effect)	Statistical Comparisons for the Significant Results	Doses (Expressed in Isoflavones)	Duration
Campbell, 2004 [15]	N = 23	Pre-menopausal (*n* = 16) and postmenopausal women (*n* = 7)	IGF-1	HDL-c: significant increase for postmenopausal women vs. placebo groups only (*p* = 0.02)	Between-group comparisons	86 mg/day or placebo	1 month
Clifton-Bligh, 2001 [16]	N = 46	Menopausal women	Unspecified in the article	HDL-c: significant increase for all doses (*p* = 0.007, *p* = 0.002, and *p* = 0.027) ApoB: significant decrease for all doses (*p* = 0.005, *p* = 0.043, and *p* = 0.007)	Within-group comparisons	28.5 mg/day, 57 mg/day or 85.5 mg/day	6 months
Hidalgo, 2005 [18]	N = 60	Menopausal women (>40 years old)	Unspecified in the article	Triglycerides: Significant decrease (p_value_ unspecified)	Between-group comparisons	80 mg/day or placebo	187 days
Howes, 2003 [14]	N = 16	Menopausal women with type 2 diabetes	Unspecified in the article	Systolic and diastolic blood pressure: significant decrease (*p* < 0.05) Vascular resistance (forearm): significant increase (*p* < 0.05)	Between-group comparisons	50 mg/day or placebo	4 weeks
Knudson Shult, 2004 [20]	N = 252	Menopausal women (45–60 years old)	HDL-c, osteocalcin, and urinary N-telopeptide	Triglycerides: significant decrease for all doses (*p* = 0.02) for 57.2 mg/day and (*p* = 0.05) for 82 mg/day	Between-group comparisons	57.2 mg/day, 82 mg/day or placebo	12 weeks
Nestel, 1999 [21]	N = 17	Menopausal women (<70 years old)	Unspecified in the article	Systemic Arterial Compliance (SAC): significant increase (*p* < 0.05)	Between-group comparisons	40 mg/day then 80 mg/day or placebo	2 periods of 5 weeks
Nestel, 2004 [22]	N = 80	46 men and 34 menopausal women (aged 45–75 years old)	LDL-c	LDL-c: significant decrease observed in men only (*p* < 0.05)	Between-group comparisons	40 mg/day or placebo	6 weeks
Squadrito, 2002 [23]	N = 60	Menopausal women (52–60 years old)	Unspecified in the article	Endothelium-dependent and flow-mediated vasodilation (FMD) of the brachial artery, ratio of oxide nitric/endothelin-1: significant improvement (*p* < 0.05 and (*p* < 0.01)	Between-group comparisons	54 mg/day or placebo	6 months
Teede, 2003 [24]	N = 80	46 men and 34 menopausal women (aged 45–75 years old)	Unspecified in the article	Systemic Arterial Compliance: significant increase (*p* = 0.04) Central Pulse Wave Velocity (PWV): significant reduction (*p* = 0.02) Effects observed for biochanin form only.	Between-group comparisons	80 mg/day (enriched in biochanin or formononetin) or placebo	15 weeks
Clifton-Bligh, 2015 [29]	N = 97	Menopausal women (average age 54 years old)	Bone Mineral Density (BMD) and LDL-c	LDL-c: significant decrease (*p* = 0.005)	Within-group comparisons	50 mg/day or placebo	>1 year
Terzic, 2012 [33]	N = 74	Menopausal women	Unspecified in the article	Total cholesterol, LDL-c, triglycerides: significant reduction (*p* < 0.001) HDL-c: significant increase (*p* < 0.001)	Between-group comparisons	40 mg/day or control (no supplementation)	18 months
Terzic, 2009 [32]	N = 40	Menopausal women (mean age 56 years old)	Unspecified in the article	Total cholesterol, LDL-c: significant reduction (*p* < 0.005) HDL-c: significant increase (*p* < 0.005)	Between-group comparisons	40 mg/day or control (no supplementation)	12 months
Chedraui 2008 [28]	N = 60	Menopausal women (>40 years old)	Unspecified in the article	Total cholesterol, LDL-c, and Lipoprotein (a): significant decrease in women with BMI ≥ 25 kg/m^2^.	Unspecified in the article	80 mg/day or placebo	90 days

## Data Availability

The data presented in this study are available on request from the corresponding author.

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
