# Peer review of "A Graded Approach for Evaluating Health Claims about Plant-Based Food Supplements: Application of a Case Study Methodology"

_nutrients, 2021, doi:10.3390/nu13082684_

Round 1

Reviewer 1 Report

TABLE 2: The title refers to the GRADE system which is not explained anywhere in the text.

LINES 100-101: It is not explained what are the current EFSA requirements and why these are not appropriate for plant-based products. Reference is made to the GRADE conditions which are not explained. It would be helpful to elaborate on this further.

LINES 124-125: These are strong wordings that would not even be considered acceptable for health claims that have already been authorised.   They should include 'contribute to'.

LINE 168-170: It does not seem appropriate to exclude studies that do not show an effect. It would only be legitimate to exclude studies that do not investigate parameters relating to the effect under consideration. The reasons for the exclusion of these studies would need to be explained better. The same in TABLE 4 and FIGURE 2.

TABLE 5: It is not clear what is meant by the title "Main criteria" and what is covered by "unspecified" as entry. If this refers to "primary parameter" this would be the correct term.

LINE 206: References 15 and 16 refer to one study, not two as the text states.

LINES 213-215: It is not clear what it means that only four studies specified key criteria.

LINE 228: What is meant by the verb "approve" in this context?

LINE 231: It would be better to replace "not representative of the general population" by which is only a subgroup of the general population".

LINE 240: "in" is missing after "change"

LINE 268: What does "composite criterion" mean?

LINES 277-278: Is this the conclusion of the case study? If so, this should be better highlighted. It does seem a quite strong message in relation to the strength of the the various studies. The statement "which helps reduce the risk of cardiovascular disease" is typical of a reduction of disease risk claim and that effect does not seem to be supported by the studies of the case study. Where does the substantiation of this effect comes from?

LINES 282-283: The relevance of the statistical data and how this affects the strength of the studies of the case study, have not been addressed in the paper. It appears that all slides have been accepted as valid and no weakness analysis was performed. This could be seen a flaw in the methodology.

LINE 300: What does 'without reducing stringency" means.

line 301: "point" instead of "points"

LINE 305: What is the relevance of this statement? Is Grade A the EFSA standard? If not, it could be helpful to illustrate that even Grade A is unachievable under the EU system for many food substances.

LINES 318-320: What does this sentence mean?

LINE 344: What does "safe information" means?

LINE 380: "parallel arms"

LINES 398-400: This sentence does not read right.

LINE 407: "is" instead of "are"

Author Response

Dear Reviewer, 

Please find in the attached document the responses to your comments. 

Thank you, 

Regards, 

Hélène Chevallier 

Reviewer 2 Report

Review on manuscript nutrients-1317274:

A graded approach for evaluating health claims about plant-based food supplements: application of a case study methodology

by Hélène Chevallier, Florent Herpin, Hélène Kergosien, Gabrielle Ventura and François-André Allaert

submitted to Nutrients

In the manuscript submitted for comments the Authors evaluated health claims about plant-based food supplements.

Generally, the manuscript is prepared correctly and fit well to the aim and scope of the journal. However before acceptance for publication in Nutrients journal the manuscript needs some revision. The topicality of the presented data raises some doubts.

Detailed recommendation:

line 5 – what means (c) near names?

lines 111-112 – is bold style needed here?

lines 132-133 – literature review should end with a clearly formulated research objective,

lines 156 – if the data was collected in December 2017, why should the publication be released only in 2021? for over three years a lot could have changed, so this raises serious doubts as to the validity of the literature review,

Tables 4 and 5 – references to literature sources should be marked with the corresponding numbers from the references section,

line 223 – incorrect citation,

lines 279-280 – incorrect citation,

line 279– incorrect citation.

Author Response

Dear Reviewer, 

Please find in the attached document the response to your comments. 

Thank you, 

Regards, 

Hélène Chevallier 
